# Agricultural Supply-Side Structural Reform and Path Optimization: Evidence from China

**DOI:** 10.3390/ijerph20010113

**Published:** 2022-12-22

**Authors:** Yun Shi, Maurice Osewe, Chebet Anastacia, Aijun Liu, Shutao Wang, Abdul Latif

**Affiliations:** 1College of Landscape & Tourism, Agricultural University of Hebei, Baoding 071001, China; 2College of Economics and Management, Nanjing Agricultural University, Nanjing 210095, China; 3Jin Shanbao Institute for Agriculture and Rural Development, Nanjing Agricultural University, 1 Weigang, Nanjing 210095, China; 4College of Land Resources, Agricultural University of Hebei, Baoding 071001, China

**Keywords:** ESG, agriculture reform, rural complex, system restructuring, technology innovation, food security, rural revitalization

## Abstract

The agricultural sector’s supply-side reform is fundamental to ensuring food security and social stability. This paper uses a comprehensive analysis method to reflect on China’s agricultural reform from 1970 to 2020. We observe that China’s agriculture made significant progress before 2020 due to preferential policies and demographic dividends. This production-oriented mode has led to the co-existence of overstocking, the rapid growth of imports, and ecological degradation. A follow-up survey acknowledged that rural complex is a comprehensive social network with substantial radiant effect involving government-sponsored projects, sector-specific programs, corporate and societal assistance. The sustainable development of the rural complex lies in industrial planning, system restructuring, and institutional arrangement. Therefore, this article anchors its system structure under the ESG principle and green development philosophy. It diversifies the agro-economy to advance digitalization and de-carbonization of the rural economy.

## 1. Introduction

China is the most populous country in the world. The well-being of people mainly depends on the ability to feed them with safe and nutritious food in a green and sustainable farming method. Besides, agriculture is the essential priority of a prosperous society and is considered a critical strategy for world economic growth and environmental sustainability. As the largest developing country globally, China uses only 7 percent of arable land and 8 percent of global freshwater to support one-fifth population in the world [1], making a great contribution to global food security. However, China’s food security faces the double pressure of domestic production and international trade. From the food production perspective, climate change will reduce grain production, raise agricultural product prices, increase net imports of most grains, and reduce China’s self-sufficiency capacity in grain. It is necessary to deepen supply and demand structural reform in the agriculture industry to achieve reasonable growth in quantity and improve quality for food security. In essence, supply and demand structural reform in the agriculture industry is a complex and systematic task involving stakeholders, such as policymakers, agricultural enterprises, family farms, and households. Moreover, it will address the challenges and threats posed by various uncertain factors.

Regarding the issues mentioned above, this study is intended to provide an in-depth overview of agricultural supply-side reform in China and propose a conceptual framework for a rural complex based on the ESG principle to optimize reform, which maybe is a research gap in current literature. We expect that this study extends the literature on this topic and provides valuable insights to guide and mold policy measures. 

## 2. Literature Review

Historically, due to the sharp increase in population after 1949, China has faced the challenges of inadequate farm produce. The decision on what is grown and in what quantities remains highly centralized [2]. As an essential component of staple food, grain is given top priority in agricultural production. The government implemented a series of incentive policies to increase grain output. The introduction of the rural household contract responsibility system in 1978 was an epoch-making initiative. It encouraged de-collectivization, clarified the most basic production relations, created farmers’ enthusiasm for grain cultivation, and liberated rural productivity, leading to a vital increase in food production and improved inhabitants’ living standards. The entry of WTO in 2001 was a critical impetus to push China to be involved in the international market. Gradually, China became a key player in global farm production policy-making. Grain production has maintained an overall balance between supply and demand since 2004 and has continuously increased in subsequent years. As a result, the Chinese economy transformed from shortage to affluent. The abolition of agricultural taxes in 2006 was a milestone in China’s agricultural reform because it alleviated the economic burden on farmers and gave subsidies directly to grain growers. This innovative measure gave rise to the boom of grain plantations. The grain output has a stable annual increase and has maintained a high degree of sufficiency. In this way, China has successfully solved the problem of food shortage and malnutrition. Data released by National Statistics Bureau in December 2020 shows that China’s total grain output increased from 113 million tons in 1949 to 669 million tons in 2020. The per capita share of grain now exceeds 474 Kg, above the international food security threshold of 400 Kg. Figure 1 shows crop harvests have kept unprecedented growth for 17 successive years. China ranks as the largest agriculture-producing country in the world. Despite notable achievements in crop production, Chinese agriculture still faces many far-reaching challenges.

The challenges include but are not limited to the low levels of marketing and organization, shorter food supply chains, low industrial income, and a rather weak international competitiveness. Besides, the spectacular rise of wealthy elites and the massive middle class have changed consumption habits from quantity-centered to quality-oriented, leading to China’s food security transformation from insufficient to structural contradictions [3]. China needs to increase food imports to satisfy an increasing demand for green and nutritional food for a healthy lifestyle. China has realized zero incidences of poverty since 2020. Nevertheless, malnutrition still exists, including micronutrient deficiencies due to unbalanced diets and a lack of vegetable and fruit intake. Specific populations such as infants, women, and the elderly face anemia and other malnutrition. The Academy of Global Food Economics and Policy (AGFEP) found that rural residents have an insufficient intake of fruit, dairy products, and aquatic products. In addition, vitamin A, vitamin C, and calcium intake are deficient among Chinese residents, especially in rural areas. Regarding diet, urban residents have the problem of slightly excessive intake, while rural residents have slightly excessive caloric intake and insufficient nutrient intake.

China experienced a continuously increasing growth rate in total imports of agricultural products faster during 2015–2020. According to the Ministry of Agriculture and Rural Areas, China’s imports of agricultural products in 2019 reached $150.97 billion, an annual increase of 10 percent. The outbreak of COVID-19 aggravated the mismatch between food supply and demand. Statistic data from Customs Administration revealed that China’s imports of agricultural products from the United States of America (US) in 2020 reached 162.74 billion CNY, an increase of 66.9%, among which soybeans, pork, and cotton increased by 56.3%, 223.8%, and 121.7%, respectively [4]. The sharp rise in food imports has highlighted the upgrading of the consumption structure and mismatched product supply. The outcome of excessive imports is that the inventory of domestic grains grows too, and over-stocking problems resulting from the government market-based purchasing policy relating to corn and rice are particularly prominent. The co-existence of over-stocking and import growth reveals that improving agricultural products’ quality standards cannot meet consumers’ satisfaction levels. Similarly, the agricultural production structure adjustment fails to catch up with the pace of integrating agricultural products’ markets and changes in comparative advantages. The most effective way is to adjust the planting structure, optimize the product mix at a fast pace, and provide diversified, high-quality products to reduce over-reliance on imports to safeguard food security better. 

Whereas profoundly influenced by the previous quantity expansion pattern of economic development, Chinese farmers concentrate on yield maximization rather than quality improvement. They got used to crop planting partly because crops have a high yield, immediate financial gains, and strong resilience to natural disaster risks. The main reason is that the government market-based purchasing policy is more elevated than the market price. Thus, it has led to the overstocking of massive agricultural commodities such as grain, cotton, and oil and insecurity of the whole industrial chain. Even worse, large amounts of fertilizer and pesticide are utilized in the production process to pursue stable and high yields, resulting in soil erosion, heavy metal contamination, and increased carbon footprint [5].

Additionally, it negatively impacts living beings’ survivability [6]. It is necessary to adjust the agricultural production structure and introduce a wide range of nutritious and resilient crops in line with the principle of comparative advantage to give full play to industrial benefits to diversify the agri-food system. This is driven by growing market demand for high-quality products. To ensure a stable food supply chain, a demand-oriented production mode should be adopted to establish a complete industrial system covering production, processing, storage, and sales. Supported by internet technology, smart agriculture is a promising option for connecting producers and consumers. In the regional and rural context, new sales models of agricultural products like community consumption, contract production, or online business will be more popular because they can help producers better access changing market information, reduce production costs effectively, and help consumers achieve their goals and specific needs. 

Urbanization is a global trend. As far as China is concerned, rapid urbanization since reform and opening in 1978 has resulted in rural-to-urban migration, loss of arable land, environmental degradation, and new dietary demands on food production [7]. Agriculture is a labor-intensive sector, and rural laborers play a crucial role in agricultural production. As urbanization progresses in recent years (see Figure 2), numerous rural laborers flock to cities to seek better pay, leaving wives, children, and parents at home to minimize living expenses. Most migrants endeavor to settle in cities to change the trajectory of their life. The Chinese agricultural employment share stood at 55 percent in 1991. It declined dramatically to 18 percent in 2017, with about 3.5 times higher value-added per worker in agriculture due to the rapid diversification of agriculture away from staples [8]. 

Equally, agricultural production is a complex and multidimensional activity carried out by producers who need to be closely integrated downstream with factor inputs and upstream with industry, distributors, retailers, and suppliers. The outflow of rural labor or migrating farmers has an essential impact on agriculture production. According to the Seventh National Census in 2020, rural people aged 60 and over accounted for 23.8 percent of the national population, 8 percent higher than the township population. The aging population in the rural area is disqualified from farming and agribusiness in medium and large-scale agriculture. Their participation in agricultural production might give rise to low labor efficiency, causing low productivity and food production insecurity [9,10]. This dilemma could derail potential economic growth.

Moreover, an imbalance in the demographic structure appears to be a significant problem, especially in rural areas. Local authorities began to switch towards machinery applications in sowing and harvesting and thus fostered mechanized service in rural areas, even in remote places, to offset the negative impact of lost rural labor. The adoption of machinery and farm mechanization services created a solid impetus for agriculture production, altered the agricultural land use pattern towards a labor-saving production mode, and supported the transformation of the agricultural land use structure in China at the expense of a sharp increase in producing costs among smallholders. 

Considering the growing loss of arable farmland, China strengthened regulations and strict control over land resources to secure a “red line” of 1.8 billion mu (120 million hectares) of arable land, spurring the overall land-use price. The agricultural production cost has been rising, posing an unbearable burden on large-scale family farms, farm cooperatives, modern agricultural enterprises, and other moderate-scaled business entities because of the ever-increasing land rentals and logistics costs. Even worse, the profit from crop planting has gone down due to changes in dietary demands. The alarming statistics are that from 2016 to 2018, data from the National Bureau of Statistics show that the average gain of three significant crops per mu is negative, i.e., rice (−80.3 CNY), corn (−12.5 CNY), and wheat (−85.6 CNY). The gap between the cost of production and income has been widening. Frustrated by rising production costs and lower profitability in crop production, most farmers resorted to the gig economy for more earnings. The lower utilization rate of farmland triggered a severe agricultural crisis and raised concerns about who will plant the crops in the future. This key point has a decisive influence on agriculture stability and potentially threatens ongoing rural revitalization [11]. In this scenario, the rural revitalization strategy designed by the China Communist party and central government is an important policy instrument to increase production, enhance rural development, maintain higher living standards, and maintain effective governance and modernization of the country’s rural areas.

Simultaneously, environmental degradation related to livestock wastes, agricultural plastics, fertilizer use, ecological contamination, unstable agricultural foundation, continued rural–urban disparities, and regional inequality remain deep-seated challenges in most rural areas [12]. 

The Chinese government has realized the negative impacts of agricultural structure imbalance. It has taken many actions for agricultural reform, destocking, reducing excessive capacity, and reducing production cost by improving the price formation mechanism of farm produce and subsidy policy. The government built a digital platform to provide information and marketing services for farmers and cultivate new agriculture business entities to shore up weak spots in the agriculture sector. These approaches have been proven conducive to bringing structural change and efficiency in resource allocation. Still, they are insufficient to solve deep-seated problems in the agriculture sector discussed in this study. Furthermore, they cannot improve the competitiveness of agricultural products and ensure farmers’ sustainable livelihood. Eventually, China determined to seek an optimized approach to deepen agricultural reform and launched the rural complex pilot construction in 2017.

The rural complex is an emergent branch linked to economic, social, and ecological components of sustainability strongly related to food safety, ecosystem management, urban–rural integration, and rural governance [13], covering modern agriculture, rural tourism, the healthcare industry, and E-commerce business and other sectors. The rural complex is still in its early stage, and its effects on agroecosystems and agricultural development are unclear. The overall system restructuring and industrial integration might serve as a powerful accelerator to optimize the farm structure. However, the rural complex is still in its infancy, and its development will likely face severe economic, ecological, and technological challenges. With the pandemic still evolving overseas, the world is confronting a scale of economic recovery and social stability. COVID-19, social conflicts, and extreme weather conditions pushed the number of people facing acute food insecurity to 155 million in 2020 [14].

Diseases and their associated problems, like hunger and poverty, are severe impediments to people’s pursuit of a better life. Regardless of effective control of COVID-19, China has to fight for national top goals in the coming years to ensure adequate grain output, agricultural product quality, and food safety for all people. Besides, China remains the world’s largest developing country, confronted by the gap between unbalanced and inadequate development, carbon peak, and carbon neutrality. People’s growing desire for a better life is challenging China’s ability to pursue high-quality products. These are the driving forces to advance Chinese agricultural transformation through agri-economy reshaping and system restructuring. This scenario raised an evident and urgent concern for agricultural supply-side reform.

## 3. Methodology

Supply-side structural reform is a complicated work involving multiple fields, which calls for trans-disciplinary research. In this paper, we have sought to complement various knowledge and information sources related to the research area, including different sectors such as agriculture, ecology, finance, tourism, and hospitality. The interaction among government authorities, rural communities, tourism operators, smallholders, farm cooperatives, and academic researchers plays a significant role in establishing a rural complex. Each participant performs their duties and jointly produces the reform results [15].

### 3.1. Systematic Literature Review

To better understand the supply-side reform of the agriculture sector, we adopted a systematic literature review (SLR) method. A systematic literature review is a comprehensive and accurate search, mainly based on specific search terms and search criteria based on the research question [16]. The paper first conducts a bibliographic search to collect antecedents on the research area. We searched out the relevant literature from the period of 2011–2021 on both Web of Science and Google Scholar by using related strings as follows (See Table 1)

From Table 1, we can see that the literature on agriculture reform is rich and fruitful, yet, many studies are not highly relevant to this paper. After deliberate literature filtering and selection, nearly 30 English documents were available for analysis. To ensure the accuracy and reliability of data, we collected multi-source data from the official website, statistical yearbook, UNFAO report, CCICED report, and government document. We also followed the inclusion and exclusion criteria, as illustrated in Table 2 below. 

### 3.2. On-Site Investigation

This paper carried out fieldwork, interviews, and conference-related activities to study: (1) On-site research on the rural complex pilot. In this paper, we selected flower and rural fruit complexes in Qianxi county, Hebei province, as a case study to illustrate the contribution to agricultural reform; (2) In-depth interviews with local authorities, agricultural entrepreneurs, owners of the family farm and rural labor, and academic researchers, focusing on the acquirement of detailed knowledge concerning reform schemes, such as policy support, development planning, industrial structure, operation and management, and off-farm employment; (3) We also conducted an academic conference highlighting training program, ecological agriculture, rural tourism, fruit processing, and agricultural digitalization. The conference’s objective was to incorporate essential knowledge from all relevant actors related to the agriculture structure reform, which is exceptionally significant regarding the rural complex’s trend mechanism and industry integration. Besides, other activities, seminars, and communications with different actors were also conducted as flexible and vital tools to facilitate the research. 

## 4. Results and Discussion

### 4.1. Deepening Agriculture Supply–Demand Structural Reform 

#### 4.1.1. Policy Support

Agricultural supply-side reform is a top-down initiative. The government plays a dominant role in establishing a supportive environment for the healthy development of the agriculture sector. In most cases, policy-making implementation highlights obstacles and sets in motion the support and infrastructure necessary for growth [17]. An agricultural policy is predominant in deepening structural reform embedded in institutional arrangement, legal models, favorable taxation, government programming, technology application, funding instruments, and practical actions. As land is a scarce and valuable resource in China and has a fixed supply, the central government implements strict regulations on the “red line” designated for permanent arable land to ensure national food security. The three-rights separation system aims to build a new agricultural management system, develop appropriate scale management, and maintain the primary stability of rural society. To increase farm productivity and resource use efficiency, governments at different levels can transform resource-consuming areas into environmentally conscious plots to enhance environmental protection, build high-standard farmland, and preserve black soil areas to control the fertility of arable land from declining. Driven by the modernization of agriculture and rural area, policymakers pay special attention to technical bottlenecks or barriers in critical fields of the agricultural sector and adopt an innovation-driven strategy to promote high-quality development. 

In recent years, climate change has increased the frequency and intensity of extreme weather events, posing a potential threat to food security [18]. To reduce the risk and associated losses as a result of natural disasters and to increase climate change resilience and adaptation strategies, concrete steps should be rolled out to ramp up subsidies for agricultural economy units. For instance, improving the minimum purchase price policy, expanding policy insurance coverage to encourage continued agricultural production, and launching educational programs to enhance risk management capability [19].

Furthermore, governments are responsible for fostering emerging industries to facilitate the development of a more stable and flexible employment market for rural labor. The potential policy instruments could increase rural income through various benefits-sharing schemes. The local authorities should introduce a policy on talent acquisition, educational programs on capacity building, and cultivating talents equipped with industrial and digital skills. Additionally, investment in infrastructure is needed to reduce carbon emissions to control pollution and build a favorable ecological environment [20]. Finally, a market-oriented, legalized, and innovative business environment is required to support rural development.

#### 4.1.2. Technology Innovation

With the increasingly intense competition in the global market, innovation is a major driving force for the sector’s growth and an effective tool to overcome technical bottlenecks impeding China’s modernization process. Agricultural innovation results from networking and interactive learning among a heterogeneous set of actors, including farmers, input industrialists, processors, traders, researchers, extension officers, government officials, and civil society organizations [21,22]. Meanwhile, it is the process through which new and exciting products and services can be used to increase effectiveness, competitiveness, resilience to shocks, and environmental sustainability. The enlisted steps could contribute to food security and nutrition, economic development, and sustainable natural resource management. It requires “a better understanding of impact pathways, new partnerships, and business models involving the public and private sectors, civil society, and farmer organizations” [23]. New technological concepts, precision, and circular agriculture, primarily inform such innovations.

Supply-side reform in agriculture is an arduous task focusing on structural optimization to accelerate coordinated and integrated ecology, economy, and society to achieve a fruitful balance between humans and nature. It requires continuous innovations in the supply chain ecosystem. China will enter a new innovation-driven stage and foster emerging business modes over the next five years. The new generation of information technology led by big data and artificial intelligence has developed fast and has become an essential economic and social transformation engine. From a macro perspective, policymakers should pay more attention to digital transformation and focus more on the 5G network, blockchain projects, and E-commerce systems [24]. The development of smart agriculture by advancing big data, cloud computing, and artificial intelligence can help farmers sell products through social networking. From the micro point of view, agricultural innovations might exist in farm practices, new technologies emergence, organizational and management techniques, social learning, and extension and advisory services. It could be, for instance, agricultural landscape, adoption of trees or cattle, forward contracting, farm tourism, diversifying the farm business, camping program, or traceability of the food chain “from field to consumer” [25].

Since the seed is of utmost importance to the yield and quality of farm produce, seed breeding technology becomes a powerful force to enhance high-quality development. To better fit into the increasing and evolving needs of food supply, developing specific cutting-edge technologies inbreeding with a solid incentive to maximize yields and quality improvements is essential.

#### 4.1.3. Industrial Integration 

According to the World Bank, a country or region will be moderately well-off when per capita GDP exceeds USD 10,000. Accordingly, the portion of leisure consumption in GDP will continuously increase because leisure attracts a lot of individual and organizational interests and promotes political, economic, social, and cultural development resulting from job creation and supply chain activities [26]. Due to rapid urbanization coupled with economic growth, China’s per capita GDP exceeded USD 10,000 in 2019 for the first time, which is a clear manifestation of the vast and multi-tiered potential for leisure consumption. In this regard, recreation and leisure will become critical elements in Chinese people’s daily lives. As a national strategy, it is vital to promote social harmony, stability, and quality of life [27].

The COVID-19 pandemic changed travel preferences from overcrowded metropolises toward spacious rural areas. Driven by the ever-increasing demand for access to nature, city dwellers, especially middle-income groups, usually view rural areas with exotic scenery, environment-friendly products, and traditional culture as “the third space” [28]. These venues provide a way to enjoy the leisure and feel nature’s healing power and the nostalgia of countryside life. The combination of relaxation, tourism, and rural life provides a solid impetus to the agri-economy that fosters catering, rural homestays, and health care services. These emergent industries are the outcome of industrial integration. On the one hand, they can promote agriculture’s transition from a singular production-oriented sector to a more vibrant, attractive, and promising sector with added value. Alternatively, they can speed up industrial restructuring, technological innovations, and social transformation.

As illustrated in Figure 3, there are two kinds of agricultural integration. Horizontal integration refers to agricultural business expansion into secondary and tertiary industries to generate new sales models, such as contracted production and e-commerce sales, which can boost new consumption in rural areas and facilitate digital agriculture. Horizontal integration is an external integration that will enhance the radiation-driving effect, accelerating the construction of a unique pattern of modern agricultural development with the ecological origin, green products, and integrated industries with highly efficient output. It will be an excellent initiative to ensure sustainable agricultural development through competitiveness [29].

Vertical integration combines diversified production, processing, packaging, sales, and distribution. It can extend and shorten the industry’s supply chain and act as value creation. In essence, this internal infusion can effectively reduce transaction costs and improve the net benefits to farmers. In this way, it can transform traditional agriculture into an advanced one in terms of a shift from quantity expansion toward quality improvement and from product-based to service-based [30]. The production structure is likely to be more scientific and comprehensive.

Regardless of the integration, both need a platform to enter different sectors and progress. A rural complex will be the right option given the scientific, systematic industrial planning and institutional design.

#### 4.1.4. Optimization Path

Agricultural supply-side reform is a comprehensive task for local administrators, agricultural entrepreneurs, smallholder farmers, and residents. It is necessary to seek an innovative approach to design a mechanism based on mutual benefits and enhanced multilateral cooperation [31,32]. In this regard, the rural complex is a new and comprehensive development model integrating all actors in a single framework. It tries to reshape modern agriculture by incorporating ecological civilization and leisure tourism across rural settings to ensure a cohesive expansion of the industries through chain extension. Summing up, the goal is to provide environmental protection and innovative technology applications coupled with traditional culture rejuvenation for the prosperity of rural people.

In light of China’s economy transitioning from high-speed growth to high-quality development, the construction of ecological civilization has entered a critical phase. Ecological restoration in rural areas tends to be more significant as it is bound to food security, life quality, and rural sustainability. Sustainability integrates environmental, social, and economic foundational facets [33]. The ESG principle, a new set of ecological, social, and governance criteria, comprising three pillars of sustainability, will be given top priority in the optimization path of agricultural reform. In the context of rural complex, E means ecosystem or environmental-friendly developing mode, S stands for social development or urban–rural integration, and G stands for rural governance regarding the institutional arrangement and structural actions of agriculture and the farmer. Under the guidance of ESG, we provide a functional framework for the rural complex (see Figure 4).

Agricultural production is the first and foremost targeted area, playing a decisive role in food supply and technological innovation of agriculture sustainability. The second is leisure creation, designed for urban residents for relaxation, recreation, and entertainment in a rural setting. The third is about the rural landscape. This area is rooted in rural resources and strives to unite terrain and farmland. It works as a value addition to rural resources through sightseeing, contributing to increased rural income. The fourth is the well-improved rural living environment. To deal with an increasingly widening urban–rural gap, a cozy and comfortable space in a rural setting where visitors and villagers can interact and enjoy leisure is appropriate.

In contrast to the conventional living area, the living environment in the rural complex requires diverse landscapes, abundant vegetation, cultural facilities, public squares, and clean, multifunctional, and beautiful streets with a better layout. The final one is the business environment. A stable and first-class business environment will provide more business opportunities for investment professionals and market entities and facilitate the mobility of capital, technology, and talent to make the rural economy more vigorous and diversified. The local administrators should use their networks to obtain resources outside the system to promote cross-departmental collaboration and co-governance [11]. This way, a stable and first-class business environment can be set up where domestic and foreign companies may develop and prosper.

The functional framework of the rural complex is not just the sum of a bunch of individual components but a whole system aimed at a healthy and friendly environment, thriving businesses, prosperity, social etiquette, and effective governance. The rural complex can achieve reorganization and collaborative development with the joint efforts of each stakeholder.

By considering rising consumption demand, the rural complex needs to integrate leisure tourism into the existing structure to unleash the potential of the rural economy. The reason for choosing agri-tourism as a primary element of the system is that it only uses a small part of the land and generates a higher return, consistent with the global principle of sustainable development [34]. Figure 5 illustrates the structure system of the rural complex and the interaction among agriculture, tourism, and community.

As a result of government interventions and market governance, the agricultural sector serves as ballast and takes full advantage of land, labor, capital, and technology. This way, we can improve resource allocation efficiency and provide quality standards in agricultural production.

Agri-tourism can be a powerful tool to diversify agri-economies and boost rural consumption. It is also a catalyst driver to encourage the two-way flow between urban and rural settings to build a unified domestic market, accelerating urban–rural integration. The rural community can contribute to the rural complex partly because it constitutes a significant part of the agricultural labor force. The rural community’s participation in tourism will build an authentic ambiance where visitors can experience true rural life. Further, specific attention should be paid to infrastructure development and public services, such as roads, sewage, education, health care, social security, and cultural inheritance, to improve the living environment for visitors and inhabitants. Through system restructuring, the rural complex has shifted its focus from a single production-based agricultural development model to an inclusive development pattern, injecting a new dynamic into the rural economy and paving the way for rural revitalization.

The rural complex comprises segmented sectors and an open, inclusive, and broad platform. Therefore, industrial integration is crucial to the future success of the rural complex. Under the direction of the ESG principle, we sketch out a multi-tier industrial cluster system of the rural complex. As shown in Figure 6, each industry plays a different role in cooperation and reciprocal relationships, generating a complex network of interconnected agricultural practices at the regional scale [35]. They share ideas, information, knowledge, rewards, and integrated risks [36] for a win-win partnership. With joint efforts of all industries, the rural complex develops into an entire industrial cluster, realizing and ensuring mutual benefits. The rural complex should apply low and zero-carbon technologies to accelerate green agricultural transformation regarding carbon peak and neutrality issues. Moreover, sustainable high-tech agriculture, like circular innovation, resource-sharing, and regenerative agriculture, can build a modern industrial network featuring a green and sustained development mode.

#### 4.1.5. Institutional Arrangement

Agricultural supply-side reform is an ongoing project. It requires strategic arrangements and institutional innovations to continuously unleash the countryside’s consumption potential and maintain sustained development of the rural economy. Agricultural innovation systems are concerned with the networks of actors from science, business, civil society, and government that co-produce the suite of technological, social, and institutional innovations that co-shape these future food systems [37,38]. Another strategy for agricultural reform to adopt is digitization. With digital technology, all the products can be grown at the best quality and reach consumers with access in just one click. By scanning the QR code with smartphones, consumers can quickly obtain information about the entire product chain through an electronic traceable encoding system. Digitization facilitates a connection between producer and consumer and makes products controllable, credible, and traceable.

Globally, a low-carbon economy is becoming a future trend. To boost de-carbonization and ecological conservation, innovation in green technology is imperative and urgent [39]. A series of stimulus policies must be rolled out on pesticide pollution, land erosion, emission reduction, ecosystem protection, and infrastructure improvement.

The healthy and sustainable development of the agricultural sector depends on policy support and capital injection. In China, policy support is related to monetary policies, regulations and insurance, operational mechanism, financial services, and land use. The capital injection includes fiscal funds, piloting social capital, and local and private investment. For instance, a national rural complex pilot can get a 70-million CNY annual special fund for system restructuring, technological innovation, and infrastructure development. Agricultural enterprises will enjoy special treatment on bank loans in R&D (Research and Development), brand building, and competitiveness enhancement. Some enterprises that hire rural labor can enjoy preferential policies on office space, funding, tax, and talent absorption.

Farmers play a critical role in the national community since they produce the food and other necessities humanity needs. Their adaptive capacity is essential to reduce the damages and take advantage of new opportunities to change current and future social and environmental stress while maintaining livelihoods [40,41,42,43]. Climate change is undoubtedly problematic for rural communities and agricultural production [44]. Governments are responsible for adopting interventions to arrest the fast-growing vagaries of the climate and carry out training sessions to build farmers’ adaptive capacity. 

Additionally, scientific and technological research about climate change enables them to respond quickly to the problems encountered in agricultural management. The loss of rural labor due to rapid urbanization and mechanization poses a significant challenge to the agro-economy and rural revitalization. Efforts are needed to encourage rural migrant workers, college undergraduates, veterans, and urban personnel to innovate and start businesses in rural areas, like contract farming or live stream.

## 5. Conclusions, Limitations, and Future Research

### 5.1. Conclusions

The study stressed the importance of Chinese agriculture supply-side structural reform and briefly introduced structural reform from 1978 until now. Further, this article outlined the political and social impacts of agricultural reform, urbanization, mechanization, digitalization, rural transition, and environmental protection. The results indicated that reform was a top-down approach, continued amid the changing situation, and made significant progress. Yet, unbalanced and inadequate development is still a big problem.

Agricultural reform is a comprehensive and multidimensional social network with the full participation of all parties from different regions, sectors, and actors. Because of the growing market demand, climate change, and uncertainty caused by the COVID-19 pandemic, supply-side reform would be propelled under the direction of the ESG principle. In this regard, a development philosophy based on an innovation-driven strategy is required to restructure the industry in a stable and orderly manner. Moreover, practical policies and institutional arrangements are incredibly significant to resource allocation, cooperation mechanisms, and agricultural transformation.

Finally, in response, this paper identified the rural complex as an optimization path to balance agricultural system structure and facilitate rural transformation. Since the rural complex is an emergent form of agri-economy, we figured out a functional framework, structural platform, and industrial cluster to maintain sustainable development.

We expect our findings to be conducive to generating a range of alternative approaches to deepen agriculture supply-side structural reform as a new development stage. It will help policymakers launch vigorous measures to advance the modernization of agriculture, encourage the flow of capital, technology, and talent in rural settings, and bridge the urban–rural development gap. Over the long haul, it will significantly contribute to a moderately prosperous society, pave the way for rural revitalization, and ensure a sufficient, stable supply of agricultural products for domestic consumption and trade.

### 5.2. Limitation

We acknowledge that this paper still leaves some room for improvement. The concept of the rural complex is progressively essential yet under-researched. It is universally recognized that talent, land, and capital are crucial to agricultural development and need a more precise presentation in the study context. Labor allocation, land consolidation, and capital input are the roots of farmers’ survival and rural development and involve various actors from different sectors. In this scenario, each issue needs an overall and detailed analysis. Likewise, there is a little elaboration on how segmented industries in rural complexes maintain shared development and achieve win-win cooperation. After all, the mechanism that needs to be designed involves the study of theoretical, empirical, and experimental systems.

### 5.3. Future Research

Further studies should seek to respond to the uncertain global environment’s dual circulation development pattern. First, alongside the rapid development of digital technologies, emerging industries and digital transformations in rural areas require in-depth research since they are critical drivers for boosting rural consumption and fueling economic growth. Second, detailed analysis ought to be carried out to solve the pain points and difficulties in product sales to reduce the production cost of enterprises and farmers. Besides, it calls for continuous and effective measures to enhance market regulation and network supervision. Another vital issue that needs further analysis is establishing a relatively fair profit-sharing mechanism between investors, enterprises, and farmers participating in the rural complex to ensure mutual benefits. Additionally, greater attention should be paid to the innovative technology in seed breeding and resource-saving to achieve high-quality economic growth and high-standard environmental protection to ensure sustainability.

## Figures and Tables

**Figure 1 ijerph-20-00113-f001:**
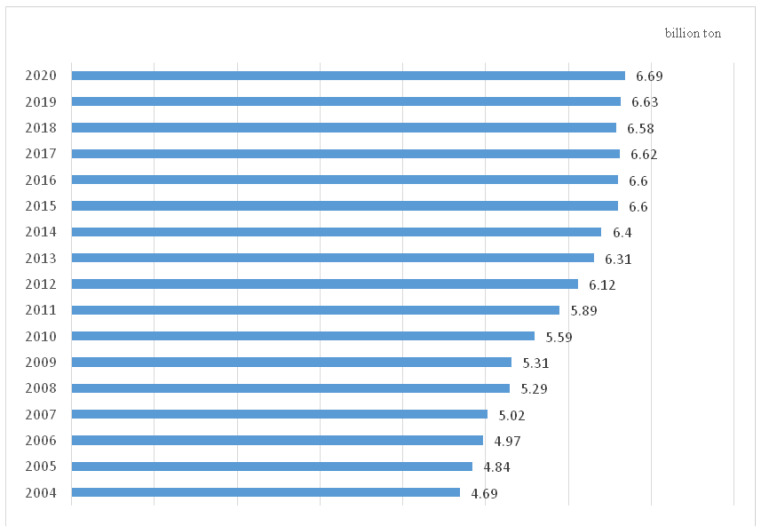
Annual grain output in China (2004–2020). Data Source: National Statistics Bureau.

**Figure 2 ijerph-20-00113-f002:**
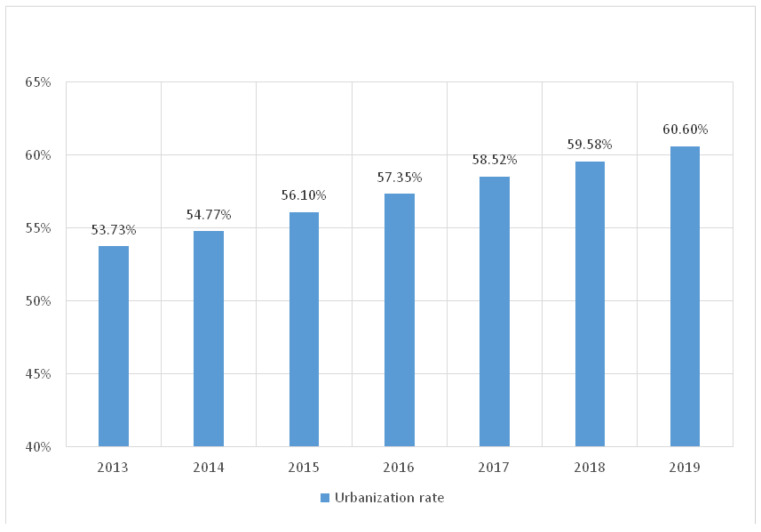
Annual urbanization rate in China (2013–2019). Data Source: National Statistics Bureau.

**Figure 3 ijerph-20-00113-f003:**
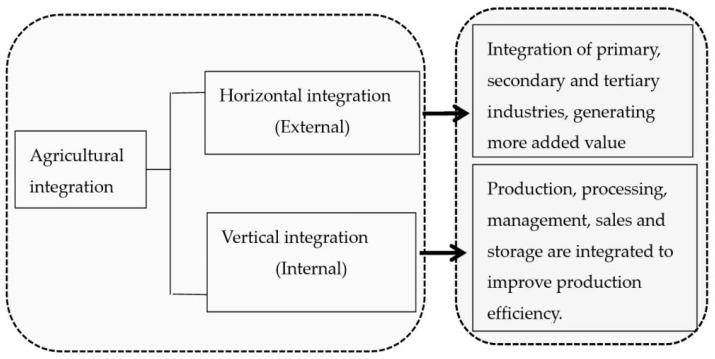
Categories of industrial agriculture integration. (Source: Drawn by the Authors).

**Figure 4 ijerph-20-00113-f004:**
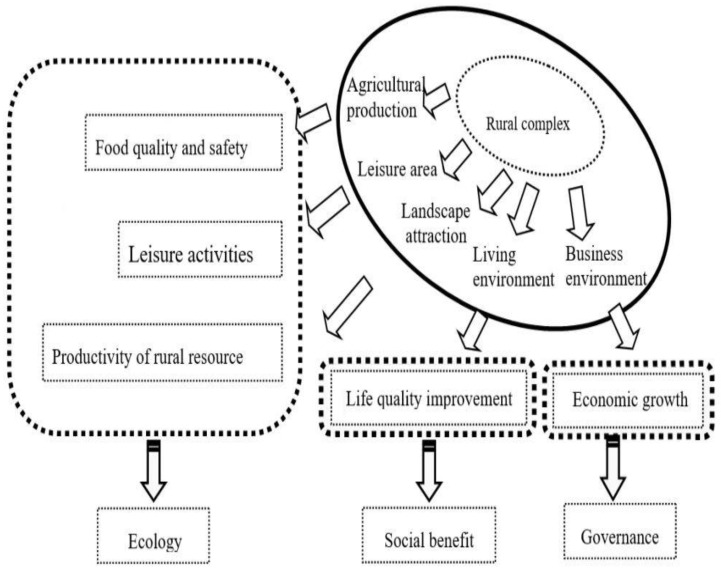
Functional framework of the rural complex. (Source: Drawn by the Authors).

**Figure 5 ijerph-20-00113-f005:**
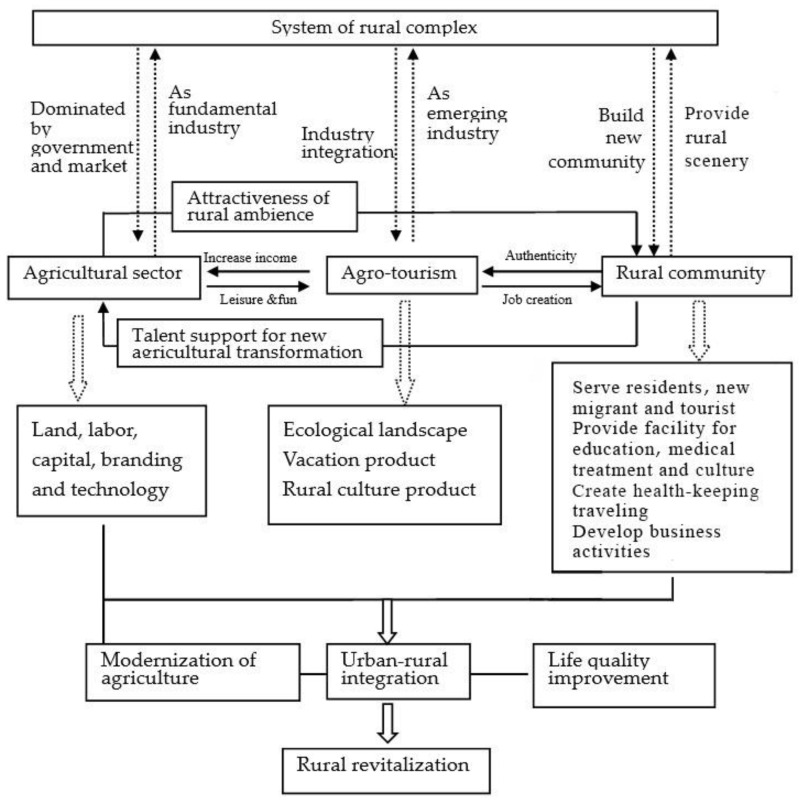
Structure platform of rural complex (Source: Drawn by the Authors).

**Figure 6 ijerph-20-00113-f006:**
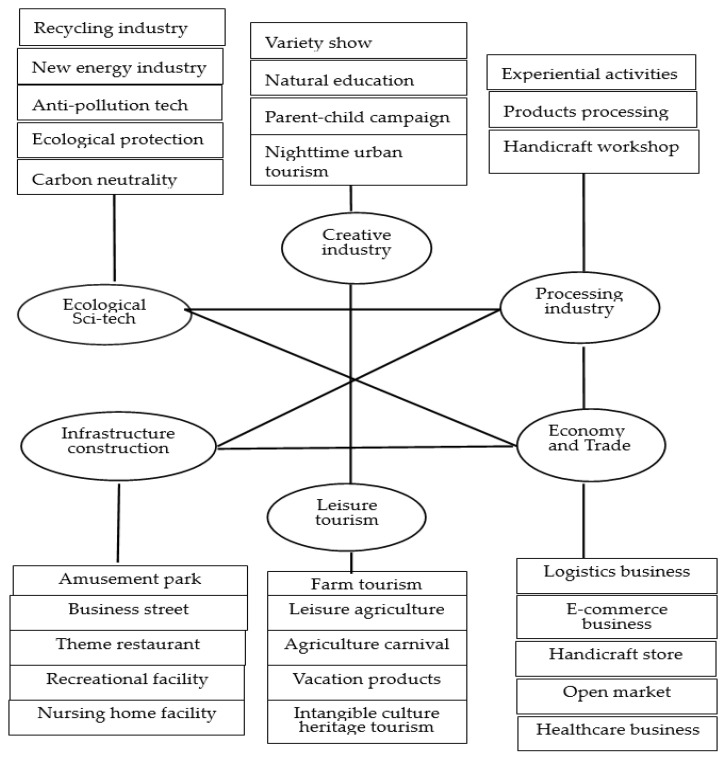
Industrial cluster framework of the rural complex. (Source: Drawn by the Authors).

**Table 1 ijerph-20-00113-t001:** The search result on supply–demand structural reform from 2011–2021.

Search Strings	Amount of Literature
Agricultural +/and structural reform +/and China	6755
Agricultural modernization +/and China	5911
Rural complexity +/and China	3548
Rural sustainability	4212
Rural revitalization	3727
Modernization of agriculture and rural areas	3193
Agricultural innovation +/and China	3604
Circular economy +/and rural +/and China	2838
Creative agriculture +/and China	521
Rural tourism	2

**Table 2 ijerph-20-00113-t002:** Inclusion and exclusion criteria.

Inclusion	Exclusion
Principal articles	Non-English articles
Articles that addressed agricultural supply, structural transformation, and optimization	Briefs and posters
Articles related to path optimization and agricultural supply-side structural reforms	Articles that address only agricultural optimization
Studies addressing limitations of path optimization and agricultural structural reforms	Articles that were not focusing on China as a case study

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
