# Peer review of "Agricultural Supply-Side Structural Reform and Path Optimization: Evidence from China"

_ijerph, 2022, doi:10.3390/ijerph20010113_

Round 1

Reviewer 1 Report

The paper reviewed on China’s agricultural reform from 1970 to 2020 using a comprehensive analysis method. I have few questions which is described below.

1. Overall good idea presented for policy implications. However it is not clear from the description that how this article is related to health-related problems? This need to be addressed in the main parts of manuscript. Also, background is too long, summarize it to one page maximum 1.5 pages. Define the objectives properly considering this comment.

2. It would be better if a pictorial overview is provided for article selection. Also, describe the inclusion and exclusion criteria. Reviewer suggest the use of PRISMA methodology for showing the selection articles.

3. The methodology needs to be divided in separate sub-sections describing search and analysis/synthesis of the articles. It will help to understand the various analysis performed on the articles as well as case analysis.

4. Result and discussion should be improved considering the first comment. Also the summary of case analysis should be discussed analytically to understand the consequences in better manner.

5. Limitations and future research should be included with the result and discussion section rather than with conclusion section.

6. Abbreviations should be expanded at first use.

Author Response

Dear Sir or Madam,

Thank you very much for your comments. Please see the attachment.

Best regards

Yours

AIJUN LIU 

Reviewer 2 Report

This research explores the supply side structural reform of China's agriculture sector. Overall, the paper has some merits, however, it also has some major issues:

1. A proper literature review section is needed to separate from the current introduction;

2. Introduction needs to be updated to provide the research rationale, motivations, research questions;

3. It is odd to include Figure 1 in the text as a main focus, while the overall research is more general;

4. Research methodology needs to be further updated with more details;

5. Section 3.2 is quite interesting, yet it cannot direct support the end of section 3.2, more details are needed.

The paper can be reconsidered after addressing the above major issues.

Author Response

(The authors gave the same response as above.)

Reviewer 3 Report

This is an interesting paper, however, in my opinio this is more "state-of-the-art" article than research one.

I am missing the proper Literature review section, gap research, RQs and hypotheses. 

The methodology should be descibed in a more detailed way. 

I am also missing the Discussion section

Author Response

(The authors gave the same response as above.)

Reviewer 4 Report

In my opinion paper quality is adequate. I just only consider that the Introduction is not the section for data presentation but it is the paper for more detailed study motivation and literature gap description.

Author Response

(The authors gave the same response as above.)

Round 2

Reviewer 2 Report

The authors have significantly addressed my last round comments. I have a minor suggestion that Table 2 can be better designed to put inclusion and exclusion criteria in parallel. 

Author Response

Dear Sir or Madam,

We appreciate the suggestions and comments you have given us to improve this manuscript. After addressing the comments, this paper is clear and significant to the readers. We have penned below our responses regarding the comment given.

Comment 1: Realigning Table 2.

Response: We much appreciate your suggestion and we have effected  it in the manuscript.

Once again, we sincerely thank you for helping us improve this manuscript.  

Yours

AIJUN LIU
